# A multivariate-driven approach disentangling the abrupt reduction of near-natural Iberian streamflow post-1980

Amar Halifa-Marín[1], Miguel A. Torres-Vázquez[1], Enrique Pravia-Sarabia[1], Marc Lemus-Cánovas[2], Pedro Jiménez-Guerrero[1], Juan Pedro Montávez[1]

[1] Regional Atmospheric Modelling (MAR) Group, Regional Campus of International Excellence Campus Mare Nostrum (CEIR), University of Murcia, 30100 Murcia, Spain.
[2] Climatology Group, Department of Geography, University of Barcelona, 08001 Barcelona, Spain.

Correspondence to: Pedro Jiménez-Guerrero (pedro.jimenezguerrero@um.es)

**Abstract**

Whereas the literature still debates how several human/natural factors contributed to the recent streamflow decline in the Iberian Peninsula, a significant abrupt decrease of Winter Precipitation (WP) has been noticed in this area since 1980s related to large-scale atmospheric drivers. This contribution assesses its potential propagation into long-term streamflow series. For this purpose, the novel dataset of Near Natural Water Inflows to Reservoirs of Spain (NENWIRES) was created. The results highlight that those higher decreases of Winter Water Inflows (WWI) are always found related to WP reductions. Whereas WP declining was strongly provoked by the enhancement of NAOi, the WWI reductions could not be essentially linked to its behaviour in several NENWIRES catchments. Instead, the intensification of permanent drought and forest extension promoted WWI reductions over the target area. In fact, these mechanisms allowed to understand why WWI reductions were higher than WP weakening. Summarizing, most humid catchments registered a WWI decline mainly promoted by the NAOi enhancement, while the extension of forest and evapotranspiration rises seem to explain the WWI losses in the semiarid environments. This contribution sheds light on the recent debate about magnitude/drivers of streamflow decline over southern European regions. Furthermore, it might help water planning with the goal of mitigating the climate change impacts affecting the water cycle.

# 1 Introduction

The Mediterranean region shows the strongest and the most consistent pattern of significant streamflow decline worldwide as a result of climate change (Gudmundsson et al., 2021). Water planning in this area thus faces the challenge of securing the sustainability of natural and human systems. Freshwater scarcity poses an incipient risk (Tramblay et al., 2020) given that intensified droughts (climate drivers) coexist with the increase of human-induced water requirements (Polade et al., 2017; Vicente-Serrano et al., 2019).

So, current studies need to shed light on the water resource variability/modelling based on multivariate approaches (e.g. Teuling et al., 2019, Massei et al., 2020).

Focusing on the Iberian Peninsula (IP), the scientific literature generally reports a streamflow decline over the last decades (e.g. Lorenzo-Lacruz et al., 2012). Nevertheless, the role of physical/anthropogenic

factors driving these reductions are still under debate.

On the one hand, the decrease of recent Winter Precipitation (WP) has been robustly reported (e.g. de Luis et al., 2010; Lorenzo-Lacruz et al., 2013). That decrease was reasonably associated to the variability of the North Atlantic Oscillation index (NAOi) (e.g. Trigo et al., 2004). Nonetheless, Guerreiro et al.

(2014) also pointed out abrupt WP decreases within the Tagus Basin since the late(early) 1970s(1980s). Other works also quantified the same drastic reduction of WP in the adjoining watersheds, Jucar and Guadalquivir (Gómez-Martínez et al., 2018, Halifa-Marín et al., 2021). They discussed 1) whether WP losses were promoted by a gradual decline or an abrupt shift; 2) the potential propagation of the WP abrupt shift have occurred into the streamflow series; and 3) its relationship with the NAOi enhancement,

phenomenon which has been widely described in the scientific literature (e.g. Luo & Gong, 2006, Wang et al., 2014). The lack of knowledge within this topic does not only concern the IP region. An abrupt change of streamflow was also reported in central/northern Europe (Hannaford et al., 2013), with a positive trend at annual/wintertime scales (e.g. Stahl et al., 2010; Vicente-Serrano et al., 2019). However, while those authors generally accepted the potential links between NAOi and the changes in wintertime

streamflow over Europe, they highlighted the importance of carrying out long record analyses in southern Europe, where data are sparse. They suggest that more long-term series are needed to determine whether

recent tendencies towards a decreased runoff in that region are found in longer records and whether this fact is related to the atmospheric circulation.

On the other hand, several contributions. have concluded that the streamflow decline was exacerbated by the temperature/evapotranspiration rise in the IP (e.g. Vicente-Serrano et al., 2014). Similar conclusions were obtained for the Mediterranean basin (García-Ruiz et al., 2011) and Europe (Teuling et al., 2019). All these works highlighted the role of a warmer climate and reforestation/afforestation processes into the evapotranspiration rise since potential evapotranspiration (ETP) also increases in response to global
warming. Peña-Angulo et al. (2021) reported that the human-induced land greening-up processes through the 20[th] century contributed to the Iberian streamflow decline (intensifying the hydrological droughts). In this line, according to Vicente-Serrano et al. (2019), human-induced land cover changes (e.g. afforestation/irrigation) mainly explain the streamflow decline in the IP. Also, the role of other impacts was assessed (e.g. the construction of dams, Lorenzo-Lacruz et al., 2012) as well as the time-lag in the
hydrological response caused by the permeability of soils (Lorenzo-Lacruz et al., 2013).

Therefore, noticeable divergences are found in the state-of-the-art regarding the importance of anthropogenic/physical drivers modulating the Iberian streamflow, which inspire a strong need for further scientific knowledge to ensure efficient water management over this target region. Also, uncertainties
persist about how the wintertime NAOi enhancement has affected the Iberian streamflow variability. As far as the authors are concerned, an abrupt change of streamflows was not fully assessed in the region. At the same time as it appears likely that NAOi enhancement provoked a sudden decrease of WP, several authors have concluded that human perturbations determined the recent streamflow behaviour instead of climate drivers.


Generally, it is well-known that hydrological response can be approximated by a simple equation (Eq. 1):

$$RR = P - E - I \,, \qquad\qquad (\text{Eq. 1})$$

where run-off ($RR$) depends on the subtraction between precipitation gains ($P$) and the accumulation of Evaporation I and Infiltration ($I$) losses.

*RR* refers to surface/underground run-off that flows outside of catchment boundaries. The estimation of evaporation is conducted by several factors (Eq. 2):

$$E = Ed + Ec + Et ,$$ (Eq. 2)

where *E* depends on the accumulation of losses due to evaporation processes from soils (*Ed*), evaporation processes from vegetation (*Ec*) and evapotranspiration (*Et*).


So, it is assumed that *E* depends on vegetation, being *Ed* higher under lower vegetation density and *Ec* increases in response to the vegetation extension. Clearly, *Et* also rises due to the extension of vegetation, but its magnitude depends on soil-moisture budgets. However, at seasonal/monthly scales, the importance of evapotranspiration prevails in the estimations of evaporation. Henceforth, vegetation changes/cover

have to be established to estimate variations in evaporation. Furthermore, it is well-known that other factors, as temperature, can impact the estimations of evaporation (i.e. through changes in evapotranspiration, *Et*). Finally, the estimation of infiltration depends on terrain characteristics such as potential water storage/recharge, permeability/porosity of soils and the moisture content, which depends on the prevailing climate conditions (e.g. precipitation records). So, as stated in Eq. 1, natural variability

of streamflow is conducted by changes in the parameters *P*, *E*, and *I*.

Under this umbrella, this work assesses the variability of wintertime Iberian streamflow in a long-term analysis (1952/2018), identifying reservoirs where their water inflows have limited human influence (near-natural environments). The relevance of wintertime changes of precipitation/streamflow resides in

its contribution to water resources, given that winter is the wet/recharge season in most areas of the IP. In addition, to the extent water planning is concerned here, the occurrence of abrupt decreases in streamflow records might severely affect the sustainability of natural/human systems (more than gradual changes). Likewise, the assessment of water resources needs to focus on near-natural catchments (e.g. Stahl et al., 2010, Hannaford et al., 2013, Vicente-Serrano et al., 2014), in order to reduce the uncertainties added by

human-induced perturbations. Because the streamflow depends on several factors with opposing effects, changes in the streamflow should be analyzed at small scales where individual factors can be understood, rather than at larger river-basin scales (e.g. Teuling et al., 2019). This approach allows to fully assess the

hydrological response to global change forcings and pressures, such as changes in atmospheric circulation patterns and/or human perturbations. In addition, the Iberian flow regimes are controlled by quite different climate conditions, which helps to assess these perturbations to a wide variety of hydroclimate areas. Therefore, this contribution aims to 1) characterize/verify post-1980 changes into near-natural water inflows series, and 2) disentangle how climate/human drivers have contributed to the magnitude of post-1980 change.

The following list of tasks has been addressed to achieve both objectives: 1) quantify the trend and the abrupt changes of water inflows series; 2) analyze whether the NAOi plays a leading role in those changes of water inflows detected; 3) identify how permanent meteorological droughts modulate water inflows changes; 4) estimate the contribution of forest extension (as a consequence of agricultural abandonment, human-induced) to the evolution of Winter Water Inflows (WWI); and 5) examine the results to provide an identification of the main precursor for WWI changes in the target catchments. For that, the initial working hypothesis is that the climate (NAOi/WP behaviour) essentially controls the reduction of water inputs, while the impact of human perturbations is weaker. The magnitude of change in any parameter of RR equation is assumed to be important to understand the variations of WWI.

## 2 Data and Methods

### 2.1 Data Sources

#### 2.1.1 Spanish Near-Natural Water Inflows to Reservoirs

In this study monthly water inflows recharging the Spanish reservoirs network (376 series) have been analysed (CEDEX, 2021). Water inflows are estimated by water management agencies from the reservoir outflow while accounting for reservoir storage changes at daily scale. For the Spanish Ministry of Environment (currently called *Ministerio para la Transición Ecológica y Reto Demográfico*) these series thus represent the official dataset of drained streamflow to the country reservoirs. Given that this analysis focuses on the variability of near-natural water inflows, data were collected from the repository using 36 series (9.6%), composing the NEar-Natural Water Inflows to REservoirs of Spain (NENWIRES) dataset.

Further details of NENWIRES reservoirs (Appendix A, Table A1) and methods used to estimate the water
inflows series are included in Appendix A.

In addition, the procedures to select the NENWIRES catchments follow commonly used methods (e.g. Hannaford et al., 2013) in this field of knowledge: 1) verification that water inflows are not affected by large perturbations, excluding those series impacted by water regulation (e.g. damming) and
urban/irrigation extractions; 2) long-term series are prioritized, and water inflows records must cover at least 46 years (70%) through the study period (1952-2018); 3) series must provide continuously records for at least 33 years (50%); and 4) records must not be reconstructed, and homogeneous quality controls have been applied to series.

The dimension of NENWIRES catchments is generally small with an average area of 929 km$^2$. The compiled data thus provides near-natural water inflows records spanning the continental Spain (Fig. 1). Likewise, the NENWIRES basins also mention the headwaters of the transboundary basins (Douro, Tagus, Guadiana), so the findings properly considered the streamflow evolution of IP. The boundaries of their drainage basins were provided by the IDE (2021).


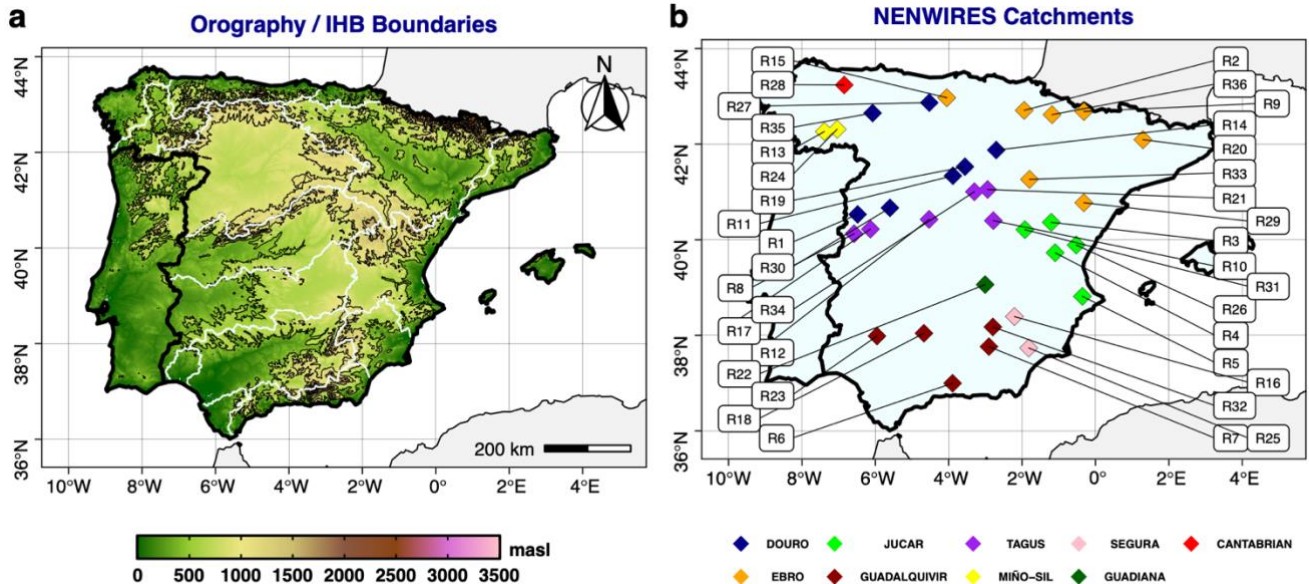

**Figure 1.** Panels show (a) the orography in the Iberian Peninsula (IP), and (b) the NENWIRES reservoirs grouped by Iberian Hydrological Basins (IHB). In the left panel, light blue contours represent the IHB boundaries in the continental Spain, and black contours mention the altitude above sea level (each 500 meters).

### 2.1.2 Climate Data

The Spanish Precipitation Gridded Dataset has been recently published by the Spanish Meteorological Agency (AEMET). The interpolation procedures manage 3,236 rain gauges. This dataset currently covers the period from January 1951 to December 2020 at daily scale. Its spatial resolution reaches ~0.05x0.05°, covering the Spanish territory except the Canary Islands. The number of records managed and its spatial resolution motivates the usage of this dataset. In the case of the Temperature Gridded Dataset of Spain, the interpolation manages 1,800 gauges, providing the daily maximum and minimum temperature. Both datasets can be downloaded in netCDF format through the AEMET Website (AEMET, 2021). Finally, the monthly NAO index (NAOi) was collected from the National Oceanic and Atmospheric Administration (NOAA), which covers the period from January 1950 to nearly real-time (NOAA, 2021). This index is a simplification of a large-scale atmospheric circulation pattern over the North Atlantic, which usefully helps to understand the principal moisture sources reaching the IP (e.g. Trigo et al., 2004).

The negative phases of the NAO index (NAOi-) are related to low systems affecting the IP, promoting moisture fluxes from the SW, whereas NAOi+ represents the opposite synoptic pattern.

### 2.1.3 Soils Permeability

Hydrological data collected from aquifers is not considered in this contribution. Instead, the groundwater modulation to WWI is estimated assessing the permeability of soils. It is assumed that this dataset helps

to estimate the behaviour of $I$ parameter into Eq. 1. The Permeability of Soils Dataset, provided by the Spanish Geological Survey (IGME), classifies soils into 9 (4) types (groups): 1) A1 and A2 represent alluvial deposits and colluvial soils formed by very permeable porous banks; 2) B1 and B2 represent carbonate bedrock, which are very permeable due to cracking or karstification processes; 3) C1, C2 and C3 represent well-drained volcanic soils, which are not common; and 4) D1 and D2 represent low

permeability and impermeable conditions, respectively. This dataset quantifies the percentage of each type of soil in relation to the basin dimension. In this study, these types of soils are classified in 4 groups: A/B1 soils are Very-High Permeable Soils (VHPS); A/B2 soils are High Permeable Soils (HPS); D1 soils are Low Permeable Soils (LPS); and D2 soils are No Permeable Soils (NPS). The dataset is online accessible through the IGME Website (IGME, 2021).

**2.1.4 Land Cover Changes**

As recommended by Teuling et al. (2019) for studies covering this topic, the HIstoric Land Dynamics Assessment (HILDA, v2.0) model reconstruction of historic land cover/use change (Fuchs et al., 2015) has been used. This dataset is based on multiple harmonized and consistent data streams including remote sensing, national inventories, aerial photographs, statistics, old encyclopedias, and historic maps to

reconstruct historic land cover. The spatial resolution reaches 1x1 km, whereas the time coverage ranges from 1900 to 2010 in decadal time steps (HILDA, 2021). The reconstruction provides information for six different land cover/use categories: forest, grassland, cropland, settlements/urban, water bodies and other. Only the changes of forest cover in the NENWIRES catchments have been quantified. The gross land changes were studied, computing the sum of all area gains and losses occurring within an area and period.

Changes in vegetation are used to estimate the $E$ parameter into Eq. 1.

## 2.2 Analysis Procedures

### 2.2.1 Study Period and Preprocessing

The study period covers from October 1951 to September 2018. Only the extended wintertime season (December to March, DJFM) is considered here, given that 1) the extended winter allows to study the contribution of snowfalls into streamflow series (early snowmelt); 2) the relation between WP and annual streamflow is intensive because the recharge of reservoirs/aquifers mainly occur at wintertime (Lorenzo-Lacruz et al., 2012); 3) WP shows the strongest and the most consistent pattern of significant decline in the IP (De Luis et al., 2010); and 3) the NAOi enhancement has been reported on wintertime. WWI generally explain more than 50% of annual records in NENWIRES catchments (Fig. 2). However, a higher contribution to annual water inputs was not confirmed within the extended winter for several NENWIRES catchments (R3, 9, 20, 21, 25, 33 and 36; 19%), where the peak of the hydrograph is driven by Mediterranean heavy rainfalls (spring/autumn) or snow accumulation/melt processes (late spring and early summer).

On the other hand, processing of NetCDF files for climate data was conducted with CDO software (Schulzweida, 2019). For converting from daily to seasonal scale (large winter, DJFM), *seassum* and *seasmean* CDO functions were used. After calculating the large winter accumulation of WP and average of maximum and minimum temperatures (WTX and WTN, respectively), its spatial average within the target watersheds is computed. This procedure was implemented in RCRAN language, loading the NetCDF files with the *brick* function (Hijmans, 2021). In addition, the catchments polygons (ESRI shapefile) were read with the *readOGR* function (Bivand, 2021). Then the gridded climate data is cropped within the boundaries of catchments using the *mask* function, and consecutively the spatial average of timesteps is estimated with the *cellStats* function (Hijmans, 2021). The extended winter accumulation/average of record series (WWI and NAOi) was also developed in RCRAN.

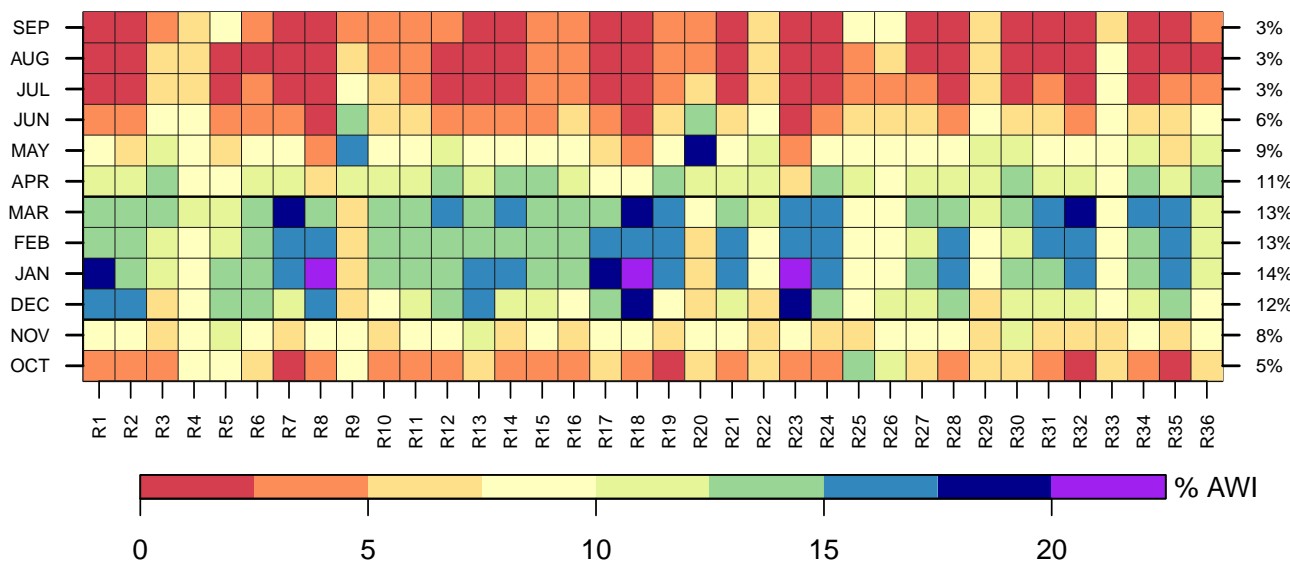

**Figure 2.** Novel hydrograph version for NENWIRES series. Each cell in the matrix shows the relative monthly contribution (rows) to the annual accumulation (AWI) in each NENWIRES catchment (columns). The monthly contribution is based on records along the study period. The monthly average of the dataset is highlighted in the right axis. The horizontal black lines highlight the extended wintertime season (DJFM).

### 2.2.2 Statistical analysis

The statistical procedures were fully conducted in RCRAN language. In order to address task 1, the trend of magnitudes for WWI throughout the study period was obtained. This trend analysis was conducted using the Sen's slope test with *sens.slope* function (Pohlert et al., 2018). To allow a proportional discussion of trend analysis, slope estimates were standardized (ZSS) as follows (Eq. 3):

$$ZSS_{WWI} = \frac{SS_{WWI}}{\bar{X}_{WWI}}, \tag{Eq. 3}$$

where standardized Sen's slope (*ZSS*) is the coefficient between Sen's Slope estimate (*SS*) and the mean WWI for study catchments.

Significant trend estimates are considered with p-value > 0.95, which is required in all statistical procedures. Once the *ZSS* magnitudes were quantified, the potential propagation of WP abrupt change to WWI since the 1980s was assessed (addressing the aim established in task 1 of this contribution). For that purpose, the most probable change was identified through the Pettitt's Homogeneity method, according to *pettitt.test* function (Pohlert, 2018). Likewise, its significance was externally evaluated with the non-parametric Mann-Whitney U test through the *wilcox_test* function (Hothorn et al., 2019). First, the breakpoint (BP) was computed for standardized series of average WWI/WP in the dataset, which were normalized through the *scale* function (R Core Team, 2021). The average WWI/WP was quantified through the *mean* function (R Core Team, 2021). A similar method was applied to both variables at catchment analysis. Then, the Relative Change (RC) of WWI/WP was quantified after the more frequent significant BP registered in the series, 1979/1980. *RC* was calculated as follows (Eq. 4):

$$RC = \frac{\overline{MS_{P2}} - \overline{MS_{P1}}}{\overline{MS_{P1}}}, \qquad \text{(Eq. 4)}$$

where *RC* is estimated by subtracting the average (horizontal bars) of WWI/WP during the first period ($MS_{P1}$) to its average during the last period ($MS_{P2}$), standardizing with the early mean ($MS_{P1}$). The average was quantified with *mean* function (R Core Team, 2021).

In order to address the task 2, the break point (BP) in the NAOi series was checked using the aforementioned methods. In addition, the correlation between WP/WWI and NAOi was also calculated. To this end, the *cor.test* function was used under Pearson's method (R Core Team, 2021), which provides its p-value. Previously, the series were detrended using the *detrend* function (Borchers, 2019). These correlation coefficients were related to the RC of WWI/WP (estimations based on Eq. 4) through the *lm* function (R Core Team, 2021). This function was used to fit linear models (regression) between variables. The function *lm* will be used afterwards in the analysis presented in Section 3. In addition, the correlation between WWI and WP was also quantified.

The abovementioned methods allow to identify those basins where WWI changes could not be explained only with the NAOi/WP variability. So, the role of several factors was analyzed (tasks 3/4). Preliminary, the modulation of WWI by the magnitude of persistent drought conditions was evaluated. The magnitude

of drought conditions is used as an estimation of moisture in soils, which impacts the hydrological response (run-off/infiltration processes) in the watersheds. To this end, the mean wintertime drought
intensity based on the previous 6/12 months was analyzed. It required the quantification Standardised Precipitation-Evapotranspiration Index (SPEI) (Berguería & Vicente-Serrano, 2017), which needs the estimation of potential evapotranspiration (ETP) using the Hargreaves method. SPEI was computed with the *spei/hargreaves* functions. These estimations allow to assess the changes of ETP in relation to the temperature evolution. The average of SPEI6/12 (through the mean function) are presented in Section 3.
Likewise, the most probable BP was also detected for those SPEI series (through the *pettitt.test* function). Finally, the absolute change of SPEI6/12 post-1980 is also shown in Section 3. The SPEI change quantified the difference between its average along 1952/1979 and 1980/2018.

In order to understand the hydrological response during the most humid winters, the potential time-lag
that permeability of soils motivates on the water generation (WWI) was evaluated in the NENWIRES catchments. This quantification allows us to understand how persistent/heavy rainfall events are linked to the recharge of aquifers, and the relationship between these modes of recharge and WWI series. Both estimations are crucial to understand the contribution of the infiltration parameter into WWI changes (parameter $I$ in Eq. 1). This analysis is of particular interest to the limestone environments where the
permeability of soils (its porosity) guarantees high infiltration rates. So, the deviation between extreme percentiles of WWI/WP in NENWIRES basins is quantified. The QQ-Deviation test (QQD) was calculated as shown in Eq. 5:

$$QQD = \overline{P_{90}(Z_{WWI})} - \overline{P_{90}(Z_{WP})}, \qquad \text{(Eq. 5)}$$

where Quantile-Quantile Deviation ($QQD$) is the difference between standardized WWI/WP anomalies
averaged (horizontal bars) to the points over $90^{th}$ percentile ($Z_{WWI}/Z_{WP}$). Higher values of $QQD$ thus show a greater water generation whether persistent/heavy rainfalls events occur (most humid winters), whereas lower (below 0) values show the opposite relationship.

In addition, in order to assesses the change of forest cover, the HILDA dataset was used. Given that this
study focuses on near-natural catchments (headwaters, depopulated areas), the changes of forest cover

were considered, as well as the agricultural abandonment and other human-induced land cover changes. HILDA provides interdecadal estimates of forest area. For our study period, a estimation per decade is available from 1950 to 2010 (7 time-steps). We thus computed the average of their interdecadal relative changes (*DRC*) as indicated in Eq. 6:

$$\overline{RC} = \frac{\sum_{i=1}^{n-1} DRC_i}{n-1} \; ; \; DRC_i = \frac{DC_{i+1} - DC_i}{DC_i} \, , \quad\quad\quad \text{(Eq. 6)}$$

where the decadal cover (*DC*) represents the forest area in each time-step, and the mean *RC* ($\overline{RC}$) is estimated by the average of interdecadal changes (*DRC*). Therefore, this method allows us to evaluate the changes of the forest area along the entire study period (shown in Fig. 5A, Appendix A).

All those methods allow to disentangle how climate/human drivers have contributed to the magnitude of post-1980 change, and to assess the contribution of abovementioned variables to estimate the principal precursor that promoted WWI changes in target catchments. Basically, a clustering methodology was applied through the K-Means algorithm (*kmeans* function, R Core Team, 2021), keeping the default algorithm settings. Once basins were classified, indicators of studied variables were computed for each
cluster. These indicators refer to the average of variables within each cluster.

## 3 Results and Discussion

### 3.1 Recent Evolution of Wintertime Iberian Near-Natural Water Inflows

The mean WWI ranged from 5 to 824 hm$^3$ in the NENWIRES basins (Fig. 3a). Higher records were observed over the western/central sector (400-800 hm$^3$), and northern/southern areas (250-400 hm$^3$). The
lower WWI were registered in the eastern/southern coast ($< 250$ hm$^3$). Meanwhile, the trends ranged from -1.8 to 0.1%/yr (Fig. 3b). 97% of WWI series have decreased in the study period. WWI only increased in one basin, but not significantly. These WWI reductions were significant in most of the catchments (55%), mainly over central and eastern sectors. Also, significant reductions frequently occurred in the most humid basins. The results thus depict significant reductions in the headwater of the Tagus, Ebro, Douro,
Segura, Jucar, and Mino basins (as defined in Fig. 1). These results agree with similar quantifications in the literature (e.g. Lorenzo-Lacruz et al., 2012), whereas this contribution is of interest to confirm the

reported changes of WWI within non-regulated catchments (Vicente-Serrano et al., 2014). Likewise, Vicente-Serrano et al. (2019) described negative trends in annual streamflows in continental Spain, which, according to these results, could be provoked by WWI changes.

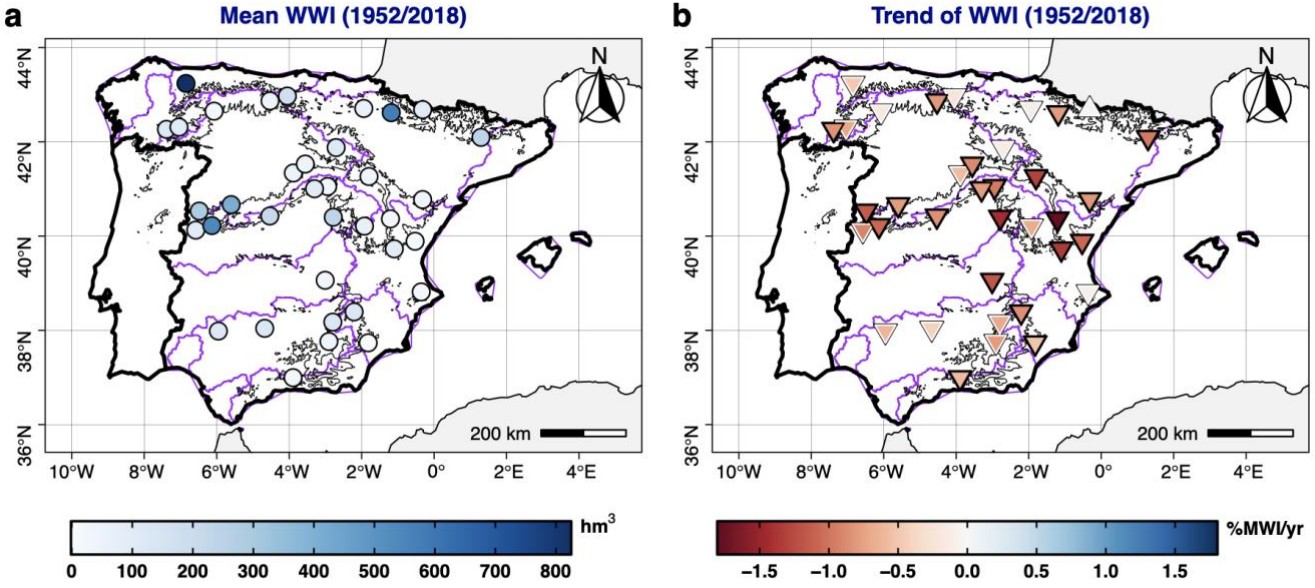

**Figure 3.** (a) Mean WWI, and (b) its Standardized Sen's slope trend estimates computed after Eq. 3 (Section 2.2.2). For the trends, symbols represent positive trend (filled triangle/point-up) and negative trend (filled triangle/point-down). In the left panel, significant estimates are indicated with a black outline. Contours of orography (black) and IHB of continental Spain (purple) were also added (see Fig. 1).

### 3.2 Abrupt changes or trend of Wintertime Water Inflows?

This section intends to shed some light to the question raised by Guerreiro et al. (2014) about whether WP gradually or suddenly changed in the IP since early 1980s. The overall negative trends of WWI could obscure an abrupt change post-1980, which has been recently noticed for the WP in the IP (e.g. Halifa-Marín et al., 2021). Thus, the most probable break point (BP) in average series of WP/WWI series (see Section 2.2.2) is identified. The average series probably are not representative for all NENWIRES catchments but give a general insight into the WWI variability.

Fig. 4 shows that WWI had a significant abrupt BP since 1979 in the NENWIRES dataset. The average

of WWI was 175.6 hm$^3$ until 1979, while it shrinks the 30% since 1980. Gómez-Martínez et al. (2018)

also identified a significant BP post-1980 on annual streamflow records in the Jucar/Turia headwaters.

The results presented here suggest that it was provoked by the abrupt decrease registered in WWI series

since 1979. Likewise, the detection of the most probable BP for WWI was also performed at catchment

scale. This analysis shows that WWI shift was detected in 1979 (41%), as well as 1978 (17%) and 1980

345  (14%).

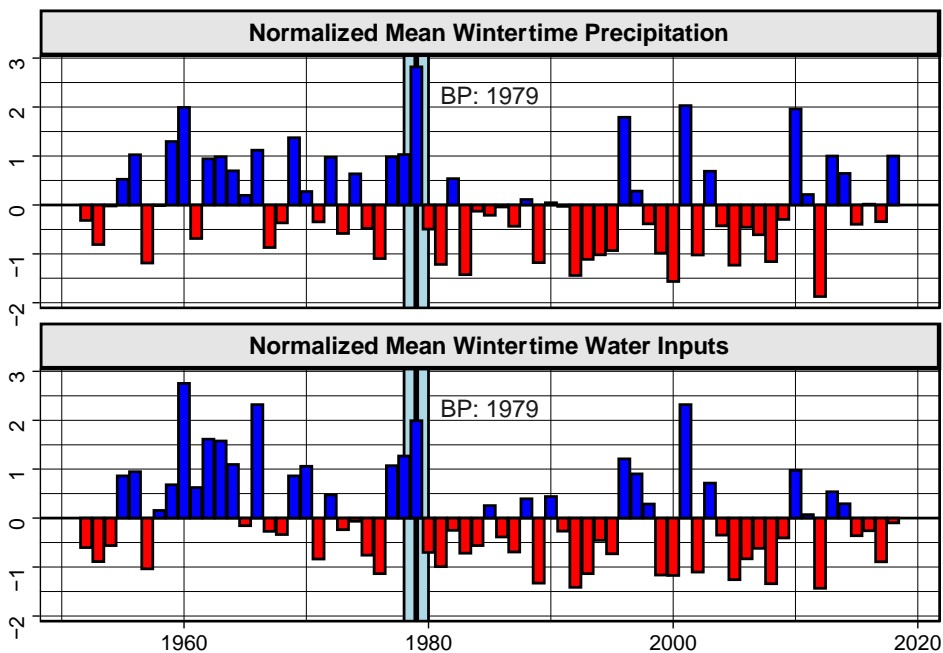

**Figure 4.** Standardized series from average Wintertime Precipitation (WP, top) and Wintertime Water

Inflows (WWI, bottom) in the NENWIRES dataset, during the period 1952/2018. Bars represent negative

(red) and positive (blue) values. Vertical blue/black lines represent the most probable BP detected through

the Pettitt's Homogeneity Test (Section 2.2.2).

Hence, a BP was found in 72% of WWI series between 1978/1980, although only 55% of these detections

were significant (Fig. 5a). We found those significant BP in the headwater of Jucar, Tajo, Segura,

Guadalquivir and Douro basins. Given that the uncertainties of the most probable change methods are

wide, the results for those basins where WWI change between 1978 and 1980 agree with the post-1980

changes in streamflows reported by Gómez-Martínez et al. (2018). Furthermore, a source of error in the BP estimation could be caused by missing values in WWI series. However, we have identified a significant BP from 1978/1980 in case of full series and series with missing values.

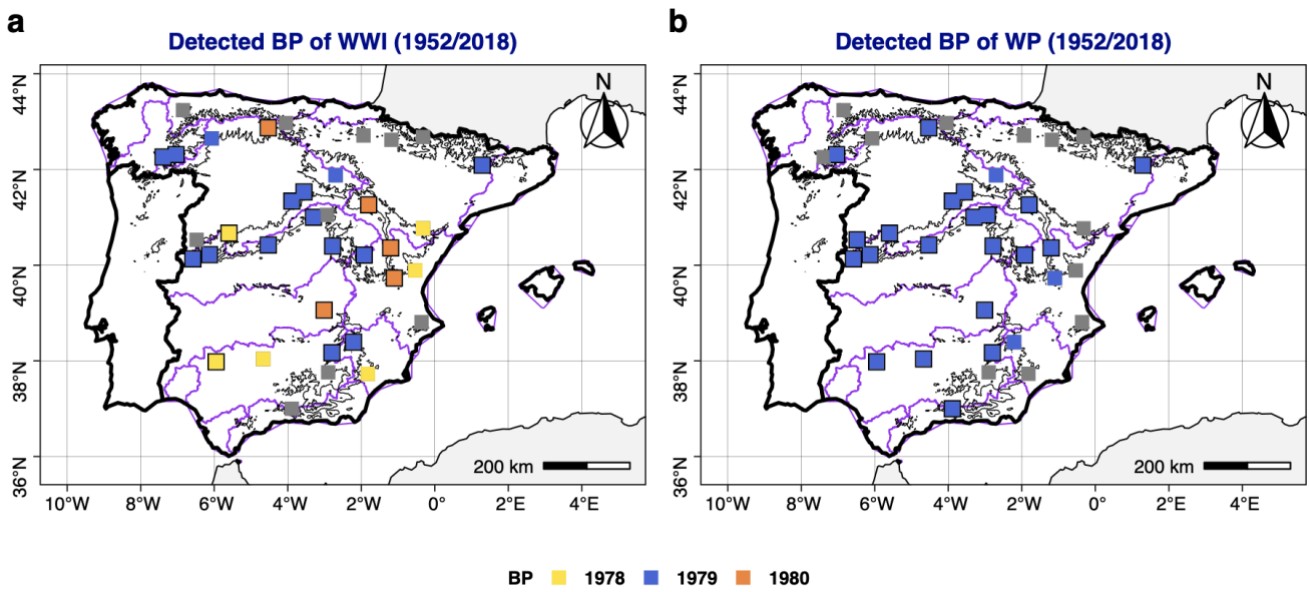


**Figure 5.** (a) Break point (BP) detected for WP series, and (b) Id. for WWI series through methods used in Fig. 4. Grey squares represent a BP not detected between 1978 and 1980. Larger squares (framed into black borders) represent the significantly BP detected in both panels. Contours of orography (black) and IHB of continental Spain (purple) were also added (see Fig. 1).


### 3.3 Does Precipitation Control the Abrupt Change of Wintertime Water Inflows?

The abovementioned references recently reported a significant abrupt decrease of WP since the 1980s across several Iberian basins, which motivates essentially this study. Once the BP has been detected in WWI series, a similar analysis was conducted for WP records. The BP detected for average WP was

found in 1979 (Fig. 4), matching the estimation for average WWI. Meanwhile, the average WP decreased by -21% since 1979, a lower reduction than the average WWI change. Furthermore, a high correlation was found between both average series (0.87). For WP, the analysis at basin scale shows that 75% of

precipitation series have a BP in 1979, which is significantly detected for 58% of the series, generally in the southern/western IP (Fig. 5b). However, the BP detection differed for eastern/northern IP. Therefore, WP generally changed in 1979, and WWI also varied between 1978 and 1980. These results agree with those of Gómez-Martínez et al. (2018) and Halifa-Marín et al. (2021), who noticed the concomitant WWI/WP abrupt changes in the headwater of southern/eastern IP since 1980, while Guerreiro et al. (2014) do not always find the BP of WP in 1979 for western IP (e.g. Tagus basin). Different time slices were analysed in all these investigations, which could explain the divergences identifying the BP in WP/WWI series. In conclusion, it seems very likely that the reduction of precipitation provoked the WWI decrease. However, the average WP decrease was more important than the corresponding decrease of average WWI.

After characterizing the BP of WWI/WP, the RC was quantified at basin scale for both variables since 1979/1980 (Fig. 6). So, the RC of WP ranged from -40% to 10% (Fig. 6a). 61% of catchments registered a significant RC of WP. A significant RC was observed in the central areas and western/southern IP. Those negative RCs also occurred where a significant post-1980 BP was not detected (i.e. northern IP). Conversely, a positive RC of WP was estimated over the eastern IP. For WWI, all catchments registered losses (Fig. 6b). The RC of WWI ranged from -60% to -3%. Major WWI decreases were observed in the headwater of Tagus, Jucar, and Segura basins (central-eastern IP, see Fig. 1). The magnitude of WWI losses is generally higher than the reductions of WP. That is, WWI also decreased where WP increased post-1980. This converse pattern suggests a poor or moderate relationship between WP and WWI in several NENWIRES basins. Vicente-Serrano et al. (2019) have already evidenced the poor relationship between climate and streamflow in basins of southern Spain. The results presented here match these conclusions, given that increases(decreases) of RC were found for WP(WWI) over southern IP. However, larger areas of Spain/IP registered WWI reductions caused by the abrupt decrease of WP (Fig. 4). Questions arise, however, about why an abrupt decrease of WP has been registered. Hence, once could wonder about the role that NAOi plays into WP changes. This question is analysed in the following subsection.

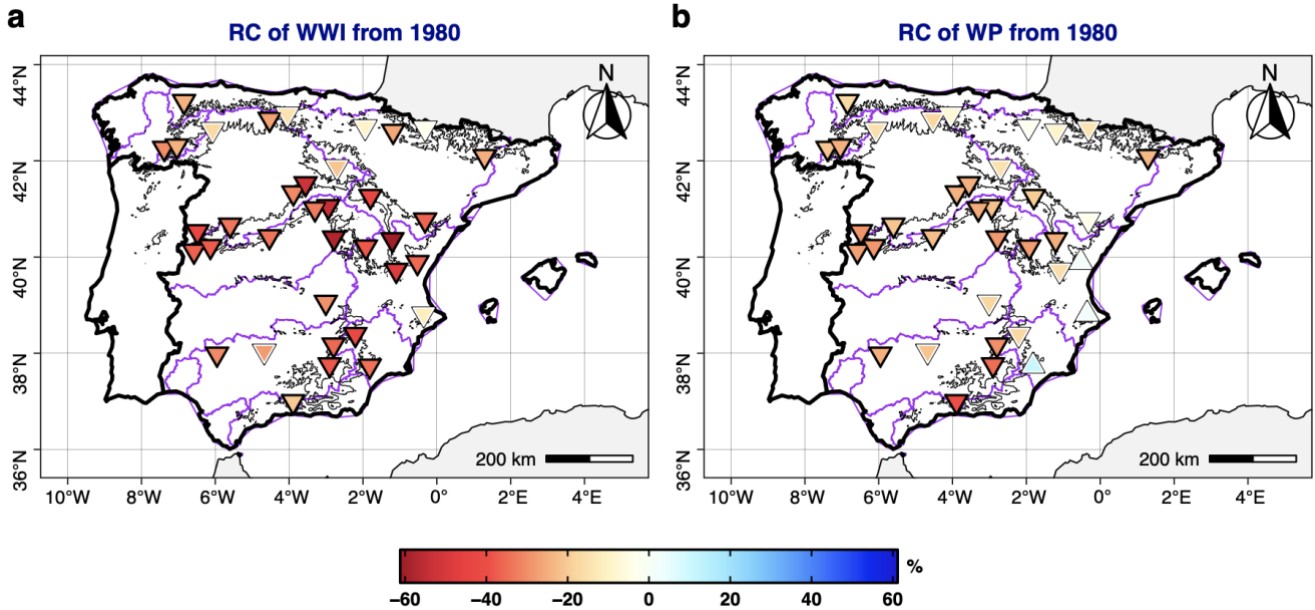

**Figure 6.** (a) Relative Change (RC) of WWI since 1979/1980; and (b) Id. for WP. The RC is computed as Eq. 4 (Section 2.2.2). Symbols represent positive RC (filled triangle point-up) and negative RC (filled triangle point down), being marked with a black outline the significant estimates. Contours of orography (black) and IHB of continental Spain (purple) were also added (see Fig. 1).

### 3.4 NAOi Enhancement Causing the Post-1980 Abrupt Decrease of Precipitation

The previous results have shown that WP/WWI suddenly decreased since early 1980s. The scientific literature has already warned about the WP abrupt decreases in several Iberian basins, which were preliminary associated to the NAOi enhancement (Guerreiro et al., 2014, Gómez-Martínez et al., 2018, Halifa-Marín et al., 2021). So, a significant BP post-1980 in NAOi series was detected in this contribution (Appendix A, Fig. A1a). Average NAOi changed from -0.35 (1952/1979) to 0.38 (1980/2018). Almost identical composites of SLP/Z500/U/V-W between NAOi-/NAOi+ phases and before/after 1980 were obtained (Appendix A, Fig. A1b-c), which confirms that winters post-1980 generally presented NAOi+ phases. A higher frequency of NAOi+ should explain the WP declining over the IP, and its propagation in WWI records, according to the literature (Trigo et al., 2004). In fact, WP/WWI (Fig. 4) and NAOi

(Appendix A, Fig. A1a) have shown a significant BP since 1979/1980. Changes post-1980 in these variables are physically coherent with the behaviour of NAO. Also, WP/WWI highly correlated with the NAOi in the NENWIRES catchments (Fig. 7). The correlation WP/NAOi ranged from -0.75 to -0.1. Generally, these correlations were significant except in the case of eastern/northern IP (see Apendix A, Fig. A2). Higher correlations were found in those basins where BPs of WP were detected in 1979 (Fig. 7b).

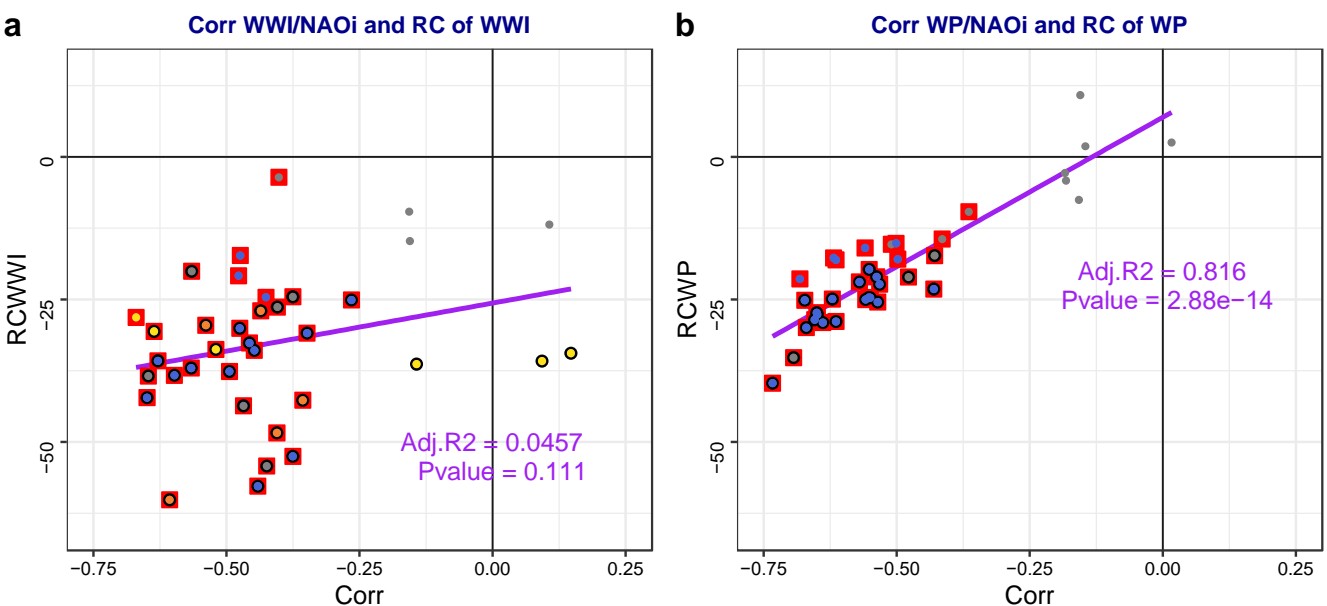

**Figure 7.** Correlation between WWI (a) and WP (b) with NAOi (X axis) in relation to the RC of each variable (Y axis). Red squares show the significant correlation values. Black circles show the significant RC of each variable. Painted spheres show the BP detected through the same rules (colour palette) used in Fig. 5. Purple line/text shows the linear regression coefficients.

Likewise, a strong relationship between the NAOi/WP correlation and RC of WP was also found. A significant adjusted $R^2$ = 0.82 was quantified through linear regression between both variables, which allows to attribute those larger decreases of WP to the NAOi post-1980 enhancement. Meanwhile, focusing on WWI/NAOi links, a similar spatial pattern (less intense) although non-significant positive correlations were observed over the eastern IP (Appendix A, Fig. A2). In this case, correlation coefficients

ranged from -0.6 to 0.2 (Fig. 7a). However, the relationship between NAOi/WP was clearly more intense than NAOi/WWI. Linear regression between WWI/NAOi correlation and RC of WWI shows a poor relationship. Therefore, the Iberian WWI abrupt reductions, understandably, depends on NAOi enhancement post-1980, whereas their magnitudes might not be essentially provoked by the WP

declining.

Likewise, NAOi shift was the principal precursor of WP decreases, especially in areas severely affected by Atlantic fronts (e.g. precipitation events coming from Atlantic Ocean, where NAOi influence is crucial to the precipitation regime). Meanwhile, uncertainties about the drivers which have promoted the NAOi

shift persist (Luo & Gong, 2006, Wang et al., 2014). If NAOi enhancement persists due to anthropogenic climate change (one of the key issues currently under discussion, e.g. Wang et al., 2014), the water resource could strongly be affected in those areas where NAOi+ increases the meteorological drought events/severity (e.g. IP, Southern Europe).

At this point, the results inspired the aim to fully understanding the causes of the post-1980 changes into WWI records, given that it has been depicted that several series were not driven by NAOi/WP changes (WWI declining shows inverse/higher magnitude than WP reductions). Also, the results of this contribution found similar divergences as noticed by state of the art literature. Whereas Gudmundsson et al. (2021) attributed the negative streamflow trends in the Mediterranean to warmer climate consequences

(e.g. decreasing precipitation, increase of temperature), Vicente-Serrano et al. (2019) inferred that human-induced factors are sometimes more important that climate for understanding streamflow trends in Spain (e.g. irrigation). Moreover, Peña-Angulo et al. (2021) conclude that changes in vegetation have a strong impact on the relationship between climatic and hydrological drought over time. Henceforth, their conclusions advice to assess the contribution of other factors to the WWI changes in the following section.

### 3.4. Propagation of Persistent Meteorological Droughts into Water Inflows Series

The SPEI6/12 index was quantified in NENWIRES basins, in order to analyse how the drought conditions might impact on WWI records (Section 2.2.2). The average of wintertime SPEI6/12 is shown in Fig. 8.

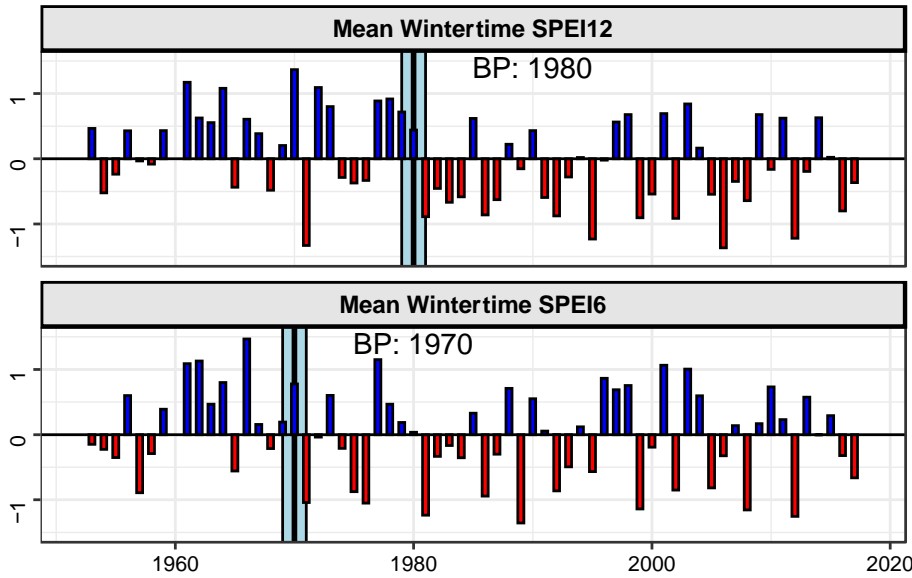

**Figure 8.** (a) Mean wintertime SPEI12 registered in the NENWIRES catchments (top) and (b) SPEI6 (bottom). Blue(red) bars represent positive(negative) estimations. The most likely change point (BP), computed through the methods used in Fig. 4, is shown for both series (light blue and dark lines).

The BP was detected in 1980 (SPEI12) and 1970 (SPEI6). Clearly negative estimations turn more frequent in both series since its BP was detected. For SPEI12, the BP detection agreed with the BP of WWI/WP series. The average SPEI12 changed from 0.37 to -0.27 since 1980. Also, persistent drought conditions prevailed in 70% (25) of post-1980 winters. These results help understanding a higher WWI decline. In those basins where SPEI12 decreases (increase of meteorological droughts conditions), WP events probably need to supply the water stress of vegetation and lower water reserves in aquifers. A simultaneous WP decline was also reported from 1980 here. In addition, SPEI12 changes also refer to the increase of ETP promoted by the temperature rise. The decrease of SPEI12 estimations in headwaters of

Spain already has been mentioned by the literature (Vicente-Serrano et al., 2014; Peña-Angulo et al., 2021). The mechanisms that could promote the droughts intensification are the decrease of cloud cover, insolation and maximum temperature rise due to NAOi enhancement. However, further work is needed to verify this assertion. In fact, an increase of maximum temperature (~0.9ºC) from 1980 is generally reported in the NENWIRES catchments (Appendix A, Fig. A3). In addition, the potential impacts on the recharge/reserve in groundwaters due to the increase of permanent meteorological droughts (SPEI12) has been addressed.

To this end, the permeability of soils was evaluated. Continental Spain shows a wide variety of soils, with permeable(impermeable) soils prevailing over central/eastern(western) IP (Appendix A, Fig. A4). The permeable soils exceeded 80% in southeastern catchments, where limestone soils are abundant. Conversely, impermeable soils reached 100% over the western IP. So, the relationship between permeability of soils and WP/WWI correlation was assessed (Fig. 9a). These correlation coefficients ranged from 0.3 to 0.9. Generally, WP/WWI are highly correlated (0.8-0.9) where impermeable soils prevail. It is well-known that correlation between WP and WWI is more intense within impermeable basins because the run-off response is instantaneous (Lorenzo-Lacruz et al., 2013). Conversely, some permeable watersheds have registered poor correlation between WP and WWI.

After WP/WWI correlation was estimated, the QQ-Deviation (QQD) (Eq. 5, Fig. 9) was quantified (the reader is referred to Section 2.2.2 for further information about QQD quantification). A significant adjusted $R^2$ above 0.35 is found in the linear regression analysis between QQD and WWI/WP correlation. So, the role of permeable soils to generate higher WWI was mostly proved during humid winters (Fig. 9a). Whereas QQD showed negative estimates over the northern/western IP (impermeable soils), positive estimates have been found in the high-permeable catchments (eastern/southern IP). The spatial pattern of QQD and WWI/WP correlation are shown in Fig. A5 of the Appendix A. These results can be summarized as follows: 1) porous watershed can infiltrate a larger volume of water; 2) Iberian extreme events of WP are characterized by the persistence of rainy days, even for several weeks; 3) which allows the water accumulation into aquifers; 4) generating an underground baseline flow joint to the surface run-off.

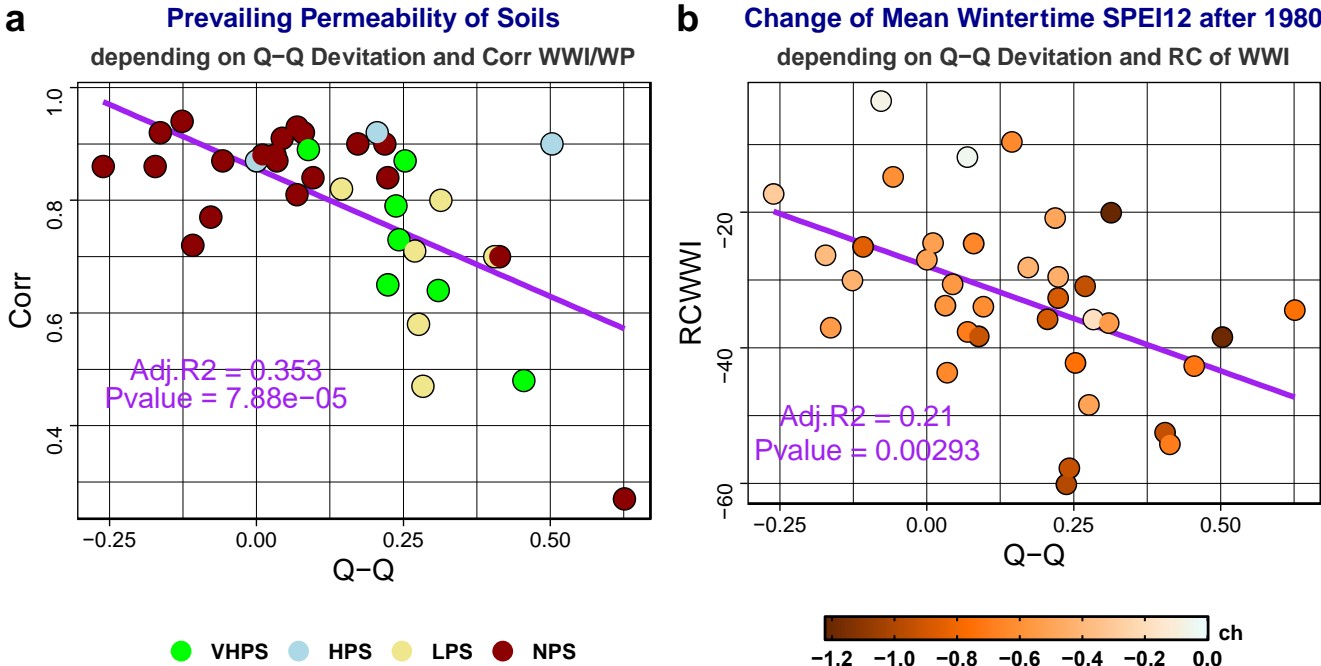

**Figure 9.** (a) Symbols represent the relationship between Q-Q Deviation (X-axis) and Correlation
estimated for WWI and WP (Y-axis), whereas are painted through the prevailing type of soils in the
catchments (permeability characteristics). Types of soils depending on its permeability conditions are
defined in Methods (Section 2.1.3). (b) Symbols represent links between Q-Q Deviation (X-axis) and RC
of WWI (Y-axis), which are painted through the absolute change of Mean Wintertime SPEI12 before/after
1980. The purple line shows the linear regression between X-Y variables, which Adj. $R^2$ and p-value are
also mentioned.

The response between WP and WWI anomalies is higher in relation to common/dry winters. This time-
lag effect probably depends on the hydrological response, water yield and the capacity of aquifers to store
water. Likewise, a significant adjusted $R^2$ above 0.2 between QQD and RC of WWI is found (Fig. 9b).
The same panel shows that the intensification post-1980 of persistent drought conditions (SPEI12) was
higher in those basins which registered higher RCs of WWI decreases and a higher QQD. These results
suggest that those basins, where the groundwater reserve highly contributes to WWI, have been affected

by the increase of droughts conditions. There, higher reductions post-1980 of WWI than WP are physically consistent because the implications of drought intensification for the hydrological response, especially under permeable conditions (where WWI also depends on groundwater reserve/flow). Future works should be devoted to further research on this assertion. In brief, the results presented here suggest that changes of infiltration ($I$ parameter in Eq. 1) have influenced the WWI reductions, given that prevailing moisture content has decreased as a result of precipitation losses at annual scale.

## 3.5 Does Land Greening-up Amplify the Water Inflows Decline?

The scientific literature has widely described that the forest extensions grew thanks to the cropland abandonment and afforestation works in Continental Spain (e.g. Peña-Angulo et al., 2021). Those authors also linked the revegetation of headwaters with an intensification of hydrological droughts. Therefore, in order to understand the higher magnitude of WWI losses than WP declining, the role of forest cover changes into the WWI variability (e.g. water generation) was further explored in the NENWIRES basins. There, the extension of forest cover was generally registered from 1950 to 2010 (Appendix A, Fig. A5b). The RC of forest cover ranged from -12% to 15%. Forest areas extended in 67% of the catchments through the study period (greening-up), whereas its cover mainly did not extent over the northern/southeastern IP. Also, the greening-up was limited where forest already exceeded 80% of watershed dimension in 1950 (Appendix A, Fig. A5a).

Gains of forest cover mostly occurred in the semiarid basins (Fig. 10a,b) where lower mean WP is recorded. Meanwhile the most humid basins registered a reduction of forest cover. This agrees to the fact that higher dimension of forest cover impacts on ETR and run-off, as shown by previous works (e.g. García-Ruiz et al., 2011, Teuling et al., 2019). So, it is reasonable to assume that the greening-up has contributed to increase the reduction of WWI in NENWIRES catchments, especially in semiarid environments. This assertion agrees with the results of Peña-Angulo et al. (2021), who confirmed the implications of forest extension in the occurrence of hydrological droughts more intense than simultaneous meteorological droughts. Those authors do not discriminate between basins based on their

precipitation regime. Therefore, the results shown here warn of the potential *Et* gains (see Eq. 1) (*E* parameter) as a consequence of the greening-up (e.g. extension of forest) that has been observed into NENWIRES catchments, which coexists with the temperature rise as a result of global warming.

In brief, one of the main results of this contribution is that a higher magnitude of WWI reductions is found
in relation to WP losses, leading to the assumption that WP was not the only cause for the WWI decline. So, basically, this study evaluated how several causes (*E* and *I* parameters in Eq. 1) have contributed to post-1980 sudden losses of Iberian WWI. To understand how each of them modulates the WWI changes, a K-Means clustering was conducted. The clustering shows catchments where WWI changes are explained by similar mechanisms/factors (Fig. 10f). 4 clusters that differed in contribution of each
precursor are identified (Appendix A, Table A2). Cluster 1 (C1) consist of 11.1% of dataset, which are well differentiated from other NENWIRES basins. These basins 1) registered the lower mean WP (164.7 mm/yr); 2) had the most important extension of forest cover; 3) registered a very poor correlation of WWI with NAOi, 4) presented a lower intensification of droughts (SPEI12 decrease); 5) had high permeable soils and 6) showed a higher magnitude of WWI reductions in relation to WP losses (-32.4%). It seems
that the WP losses and forest extension drove the higher magnitude of WWI reductions in those basins.

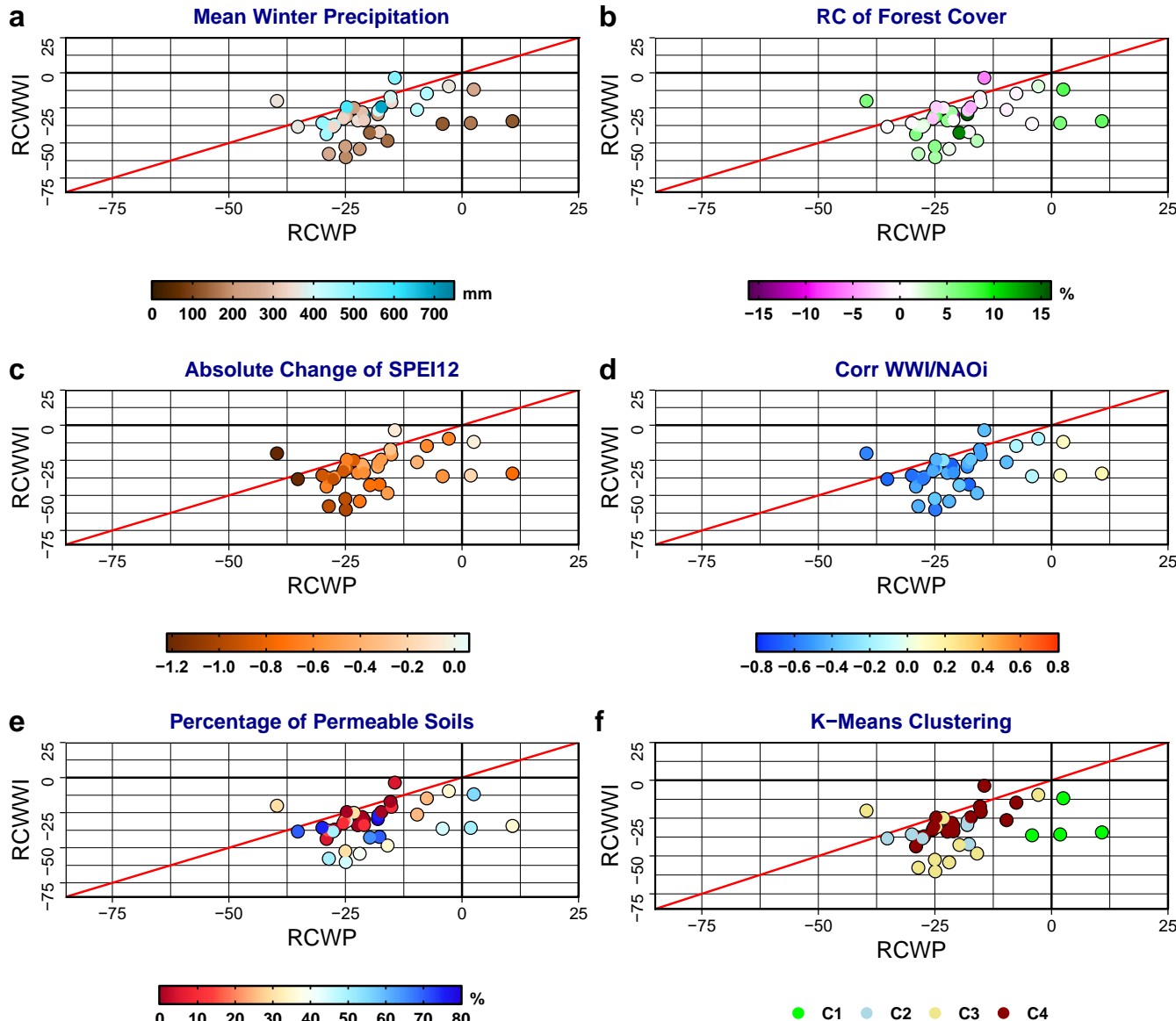

**Figure 10.** Scatter plot of relative changes of WP (X axis) and WWI (Y axes). Symbols are filled by the magnitude of several variables: (a) mean WP; (b) RC of forest cover computed after Eq. 6; (c) absolute change of SPEI12, (d) correlation between WWI and NAOi; (e) percentage of permeable soils; and (f) K-Means clustering depending on abovementioned variables.

The ETP is assumed to increase due to forest extension, while SPEI12 quantifications did not intensify since 1980. So, although the WP does not change significantly, WWI reduced by -29.6% because the rise of outputs affecting the water yield. However, hydrological modelling is needed to confirm this assertion.


Cluster 2 (C2) covers the 16.7% of basins where 1) mean WP is 355.2 mm/yr; 2) permeable soils are abundant (70.1%); 3) WWI highly correlated with NAOi; 4) SPEI12 severely decreased; 5) forest cover extended 2.5%; and 6) WWI reduced more than WP (-10.8%). WWI changes are seemingly provoked by NAOi enhancement, whereas the magnitude of reduction was intensified by the amplification of permanent meteorological droughts since 1980, which affected to the groundwater reserves, given that C2 consist of, generally, permeable soils. Clearly, the extension of forest also contributes to reduce WWI, as suggested for C1 basins, but in the case of C2, forest extension was less important.


Likewise, Cluster 3 (C3) includes 27.7% of catchments where 1) mean WP is 232.4 mm/yr; 2) permeable soils are less frequent (40.5%); 3) WWI correlated with NAOi (-0.4); 4) SPEI12 decreased -0.9 since 1980; 5) the extension of forest cover was important (4.7%); and 6) WWI reduced more than WP (-17.5%). In this group of basins, the higher reduction of WWI is driven by the same processes as for C2, but C3 registered higher intensification of SPEI12 (duration of meteorological droughts) and extension of forest. Oppositely, the influence of NAOi is less important for C3. Also, the RC of WP was weaker.



Finally, Cluster 4 (C4) covers 44.4% of basins. Most humid basins are grouped into C4, where 1) mean WP is 445.2 mm/yr; 2) soils are mainly not permeable; 3) WWI highly correlated with NAOi (-0.5); 4) SPEI12 decreased -0.5; 5) forest cover does not change (-0.1); and 6) WWI changed more than WP (-7.1), however the magnitude of change was almost similar for both variables. Clearly, in the C4 basins, WWI changes were driven by NAOi enhancement and amplification of meteorological droughts. Likewise, given that C4 basin are generally impermeable, the groundwater reserve cannot supply the decrease of surface run-off during dry events.


## 4. Final Remarks

This contribution focused mainly on disentangling the abrupt WWI reduction post-1980, echoing the call made in Halifa-Marín et al. (2021), for assessing the potential propagation of abrupt changes in WP records into streamflow records. Therefore, the main findings of this contribution can be summarized as:

1) The NENWIRES dataset was created to analyze the recent evolution of near-natural Iberian streamflow draining to reservoirs in headwater catchments (NENWIRES). We identify a significant reduction of the WWI, that is related to an abrupt change after 1980 for most of basins. This change agrees with the abrupt change reported for WP by other authors and confirmed in this contribution. This allows to analyze the changes reported here as the differences between two periods (1952/1979-1980/2018).

2) The decrease of WP is the main driver of the decline of WWI. However, a higher magnitude of WWI losses rather than WP reductions has been generally quantified. These extra water losses depend on the extension of vegetation (*E* parameter in Eq. 1), and moisture content of soils (e.g. its permeability characteristics, *I* parameter in Eq. 1), which, in turn, also differed between the variety of climate conditions (e.g. precipitation regime).

3) A strong reduction of WWI has been observed in a set of semiarid catchments, where WP slightly increased, and where the NAOi does not exert an important influence. Those basins (grouped in the C1 cluster) are characterized by permeable soils, which are affected by the most important extension of vegetation. So, it is assumed that this greening-up in semiarid environments, where water storage of soils is abundant, has provoked significant evapotranspiration rises. Consequently, those water losses have decreased the WWI records. Similar assertions are concluded for catchments classified in the C3 cluster, where, however, WP is promoted by the NAOi influence, although they also are semiarid environments. C3 catchments registered the higher decreases of WWI.

4) Another set of most humid catchments (C4 cluster) is characterized by impermeable soils, and unchanged vegetation cover. There, the magnitude of WWI and WP reductions is almost similar. Meanwhile, the WP of catchments grouped into C2 cluster is characterized by a strong influence of the NAOi, and the extension of vegetation. Then, higher reductions of WWI are observed in relation to C4 catchments.


These assertions allow to conclude that a higher mean precipitation induces a minor role of evapotranspiration/infiltration losses into streamflow changes. Nonetheless, water generation in semiarid catchments is widely affected by those losses, which can play the main role, as important as the decrease of precipitation. In terms of Eq. 1, in case of lower values of precipitation ($P$), the importance of $Et$
(vegetation cover) and $I$ (water storage) is large. When $P$ is higher, $RR$ is almost similar as $P$, as well as when $E$ and $I$ are weaker.

Last, the conclusions to this contribution confirm that the initial hypothesis was not accurate, since WWI changes essentially do not depend on climate in the set of NENWIRES catchments. Abrupt changes do
not impact on water planning in the same way as gradual changes, so that policies adapted to this mode of climate variability could be necessary. Future works will have to deal with the improvement of the scientific knowledge about shifts of the NAOi and their implications to hydrological resources in southern Europe. The findings presented here thus encourage the need to develop a deeper knowledge about NAOi variability under a warmer climate and conduct high-resolution modelling considering the water losses
because of vegetation extension and moisture-soil content, especially in semiarid environments where the availability of freshwater is crucial. Henceforth, this work can contribute to the mitigation of warmer climate impacts on water cycle/planning in the Mediterranean area, the IP or other global semiarid environments.

*Data availability.* All data sets used in the current study are publicly available from the indicated references or sources (See assets in the doi of manuscript). Also, we will be pleased to send the NENWIRES dataset under request.

*Author contributions.* A.H-M conceived the original idea and designed the overall study. A.H-M, E.P-S and M.T-V developed the NENWIRES dataset. A.H-M, and JP.M performed the analysis. All co-authors contributed to the interpretation of the results. A.H-M led the writing of the paper, with contributions of P.J-G and JP.M.

*Competing interests.* The authors declare that they have no conflict of interest.

*Acknowledgements.* The authors thank the reviewers and the editor of the manuscript for their valuable contributions and fruitful discussions.

*Funding.* The authors acknowledge the ECCE project (PID2020-115693RB-I00) of *Ministerio de Ciencia e Innovación/Agencia Estatal de Investigación* (MCIN/AEI/10.13039/501100011033/). A.H-M thanks his predoctoral contract FPU18/00824 to the *Ministerio de Ciencia, Innovación y Universidades* of Spain. E.P-S thanks for his predoctoral contract to the ACEX project. M.L-C thanks his predoctoral contract FPU17/02166 to the *Ministerio de Ciencia, Innovación y Universidades* of Spain.

*Review statement.* This paper was edited by Erwin Zehe and reviewed by A.J. Teuling and one anonymous referee.

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
