# Peer review of "A multivariate-driven approach disentangling the abrupt reduction of near-natural Iberian streamflow post-1980"

_Hydrology and Earth System Sciences, 2021_

## Author Response (AR1)

Response to editor/referee comments on

**A multivariate-driven approach disentangling the abrupt reduction of near-natural Iberian streamflow post-1980**

Amar Halifa-Marín[1], Miguel A. Torres-Vázquez[1], Enrique Pravia-Sarabia[1], Marc Lemus-Cánovas[2], Pedro Jiménez-Guerrero[1], Juan Pedro Montávez[1]

[1] Regional Atmospheric Modelling (MAR) Group, Regional Campus of International Excellence Campus Mare Nostrum (CEIR), University of Murcia, 30100 Murcia, Spain.
[2] Climatology Group, Department of Geography, University of Barcelona, 08001 Barcelona, Spain.

Correspondence to: Pedro Jiménez-Guerrero (pedro.jimenezguerrero@um.es)

All the responses to referee comments that were uploaded as comments in the interactive discussion have been introduced in the new manuscript version. The discussions authors considered interesting for the article understanding have also been included. A changes control version is included along with the new manuscript version. Thank you very much to both referees and editor for their contribution to the significant improvement of the manuscript robustness and content quality with their suggestions and comments.

---

## Author Response (AR2)

Response to editor/referee comments on

**A multivariate-driven approach disentangling the reduction of near-natural Iberian streamflow post-1980**

Amar Halifa-Marín[1], Miguel A. Torres-Vázquez[1], Enrique Pravia-Sarabia[1], Marc Lemus-Canovas[2], Pedro Jiménez-Guerrero[1], Juan Pedro Montávez[1]

[1] Regional Atmospheric Modelling (MAR) Group, Regional Campus of International Excellence Campus Mare Nostrum (CEIR), University of Murcia, 30100 Murcia, Spain.
[2] Andorra Research + Innovation, Rocafort 21-23, AD600, Sant Julià De Lòria, Andorra

Correspondence to: Pedro Jiménez-Guerrero (pedro.jimenezguerrero@um.es)

We welcome the feedback of both referees and appreciate all the suggestions and comments. The majority of them have been included in the final manuscript version.

**Guidelines**

Referee comment in black.

Authors answer in blue.

**Answer to the Reviewer 1 comments**

**Regarding the title.** I do not think we can speak of an "abrupt" reduction of the streamflow after 1980. The decrease must be considered in a wider context and definitively, the suggested phenomenon of streamflow decrease is not starting in the decade of 1980. There is literature that support a progressive reduction as consequence of different factors. I suggest the following title: "A multivariate-driven approach disentangling the reduction of near-natural Iberian streamflow".

- Title of the work has changed. The reference to "abrupt" has been removed. Thanks for the suggestion.

**Line 15.** Again. The decrease of rainfall can be considered in a wider temporal context. See recent IPCC report (Chapters 11 and Atlas) or Peña-Angulo et al. 2021 Env. Res Lett. Short-term precipitation trends are mostly related to the natural variability. Thus, the recent IPCC report (see Atlas chapter) shows an increase of precipitation in the region since 1980s. The authors should stress the importance of these "SHORT-TERM" precipitation trends to explain availability of water resources, but I would not state "abrupt" trends as common pattern of precipitation in the Mediterranean region is the strong interannual and decadal variability.

- We have replaced "abrupt decrease since 1980s" by "continuing decrease during the second half of 20th Century".

**Line 22.** What is permanent drought? Drought is a temporal anomaly regarding long-term conditions. This statement should be removed.

- The "permanent drought" was replaced by "drought conditions" in the manuscript.

**Line 20.** Better "enhancement of the NAOi during the study period". The recent IPCC report (AR6) stresses no temporal long-term changes for the NAO and main control of natural variability. The way in which this is written suggests certain role of Climate Change on NAO trends that should be omitted in the ms.

- Thanks for the recommendation, the clarification "during the study period" has been added.

**Line 31.** The climate change attribution should be removed here. Land use changes and irrigation demands are main drivers of streamflow reduction in the region. See Vicente-Serrano et al. Geophys Res Lett. 2019 and 2021, Peña-Angulo et al. 2020 Antrophocene and the several references cited in these studies. Note that the manuscript focuses on the winter season but trend analysis in these studies focus on drought events across different seasons and on annual streamflow.

- Thanks for the recommendation. That sentence has been removed.

**Lines 42-50.** Remove the reference to precipitation trends and stress the role of the temporal variability of precipitation in the Mediterranean. There are not evidences of long-term trends in precipitation in the region. See Peña-Angulo et al. 2020 Env. Res. Lett. And IPCC AR6 Chapters 11 and Atlas.

- Thanks, this paragraph has been modified.

**Lines 60-70.** This is a great point. The focus of the previous paragraphs should stress that although long-term trends in precip. are not evident, short-term precipitation trends as observed between 1960s-2000s associated to a positive phase of the NAO could strongly determine the availability of water resources in recent periods as consequence of the higher influence of other factors as land cover changes, temperature increase, irrigation demands, etc.

- A larger discussion of the WP declining, in relation to the precipitation variability since 1850, has been included.

**Line 76:** Again, I would not speak of an abrupt change…

- In this sentence "They discussed 1) whether WP losses were promoted by a continuing or an abrupt decline", we are citing the article of Guerreiro et al.

(2014), where they precisely purposed this question due to the precipitation variability detected in the Tagus Basin. In the end, they compared their results of trend and homogeneity in series. A quite similar methodology which we use in this contribution. That's why we consider mentioning "abrupt".

**Line 75:** NAOi enhancement by NAOi variability

- A clarification is added, "its relationship with the NAOi enhancement during the last decades of the 20th Century"

**Line 77:** Better: "NAOi variability may cause strong decrease of WP in particular periods"

- This change has included in the text.

**Lines 80-100:** These paragraphs are losing the focus. This information should be moved to the section of methods.

- The paragraph explaining the Eq. 1 and Eq. 2 has been removed, also pleasing the comments of the reviewer 2.

**Line 101:** Replace: "identifying reservoirs where their water inflows have limited human influence (near-natural environments)." By "in near-natural catchments".

- Done, thanks.

**Line 106:** "abrupt decreases" by "short term decreases"

- Done. This suggestion has been included in the manuscript.

**Line 114:** redefine the objectives: 1) characterize recent streamflow changes in near-natural catchments of Spain and 2) disentangle how climate/human drivers have contributed to the magnitude of the changes.

- Thanks for redefine the objectives. This section has been modified.

**Line 121:** Again "abrupt": "short term variability and changes"

- Thanks. It has been changed.

**Line 122:** Remove any reference to "permanent" droughts throughout the entire manuscript. This is not a correct terminology.

- Thanks, the mention of "permanent" has been removed in the manuscript.

**Line 124:** Note that vegetation is mostly active in the warm season so a role of vegetation trends on streamflow should not be expected in the winter season. See Vicente-Serrano et al. 2021 Geophysical Research Letters.

- Thanks, this suggestion is included in the new version of the manuscript.

**Line 127:** Rewrite: "controls the variability and short-term trends of winter streamflow, while the impact of human perturbations is weaker in winter". If the focus is exclusively on winter streamflow this is evident as water uses by vegetation and irrigation are small in this season and the variability/trends in streamflow are mostly determined by precipitation. Also note the small role of temperature increase as it mostly affects streamflow as consequence of enhanced water demands by natural vegetation and irrigation in summer months.

- Done, this suggestion is included in the new version of the manuscript.

**Line 145:** You note that the start or end period of the series could affect trends. This is always problematic to compare series that start/end in different years.

- Thanks, but we prefer to use the data how the official institution provide it. Even if we assume some uncertainties due to the use of series including missing data. However, the majority of series do not have lack of records.

**Line 204.** This should be explained earlier in the introduction. The exclusive focus on winter is controlling the obtained conclusions and interpretations. The authors should stress that some of the discussion (e.g. precipitation vs. temperature increase and human attribution) is affected by different seasonal mechanisms. The authors should stress that whereas winter precipitation may be the main driver of winter streamflow, other mechanisms can be more relevant in the warm season and they are not analysed here. The focus on winter season can be perfectly valid and it can be justified given relevance on reservoir storages, hydropower production, etc., but the discussion on the contribution of precipitation vs. human uses is merging different mechanisms that are playing a very different role on different seasons.

- Done. Thanks.

**Line 240.** P < 0.05?

- Right, thanks.

**Lines 241-264:** Are these methods necessary to assess the contribution of precipitation and NAO to precipitation? A simple regression analysis between precipitation, NAO and streamflow and temporal analysis of the residuals could be definitively less complex and more clear (See Beguería et al. 2003 Ambio).

- We think that it is necessary to achieve the objective of this contribution, given we try to evaluate the WP/WWI declining (i.e. abrupt or gradual). Assuming that the intensity and duration of changes can be crucial in the magnitude of impacts on human/natural systems We hypothesise that if WP/WWI registered the same sudden change (at the same time), it could be an estimator of the relationship between variables.

**Lines 267-277:** Why is not previous summer/autumn streamflow included as a better metric of the water depletion in the basins instead of a drought index? I think this would provide a better metric of the water availability in the basin prior winter and thus, it could be included in the analysis as independent metric. This analysis could produce very interesting results: i.e. if summer and autumn water availability reduces as consequence

of enhanced consumption by plants (given natural revegetation and warming), it could be possible to check its role on winter streamflow and to separate between the possible role of precipitation variability and trends and the role of the trends in water consumption over the warm season.

- We focused on wintertime changes of the variables. Of course, we assume the propagation of changes occurring in the rest of seasons, and it is true that the vegetation activity is weaker on winter. In the new version of the manuscript, we stress these important issues, including the Relative Changes of water inflow and precipitation in spring, summer, and autumn.

**Lines 305-306.** Remove reference to post-1980 change and stress temporal variability and short term changes.

- Done.

**Figure 4.** Series are not correctly normalized. Magnitude of positive values is higher than of negative values. Simply use the correct probability distribution to normalise the winter precipitation and streamflow series. It is not possible to use the mean and the standard deviation as the series are not normal. I do not think it is correct to state "abrupt" changes in these variables without considering longer series. See e.g. Peña-Angulo et al. 2020 Env Res Lett and Vicente-Serrano et al. 2020 Int. J. Clim. I would remove the vertical bar and the mention to abrupt changes.

- Thanks. We have removed the plot with standardised series (now it is included the percentile rank of series). We have clarified that 1) 'abrupt' only refers to strong change between two periods, a significant breakpoint found in the series, and 2) it is just concerning the study period.

**350-365:** remove this analysis and Figure 5. Not necessary and debatable interpretation.

- Done, thanks.

**Replace by:** 3.3 Does Precipitation Control the variability and short-term change of Wintertime Water Inflows? Reinterpret this section removing the reference to "abrupt" changes.

- Done, thanks.

**Figure 6.** Include trend analysis from the beginning of the study period, not only from 1980.

- Post-1980 mentions the change between the first period (1952-1979) and the last one (1980-2018). It has been clarified in the manuscript.

**455-460.** Note again the strongly relevant assessment of the seasonal drivers of streamflow variability and trends. You cannot merge in the interpretation processes that are playing a role on different seasons. Vegetation is non-active in winter and water consumption is small so it is not expected a large contribution in winter streamflow. This should be stressed in the ms.

- Thanks, that is a very good suggestion for the best improve of this work. Now, the manuscript notices these issues in the discussion.

**Section 3.4.** It would be very interesting to include summer/spring streamflow trends as precursor of winter streamflow variability. This would allow to determine if winter streamflow is affected by higher or lower water consumption by natural vegetation associated to revegetation processes and warming trends.

- The Relative change of both variables during these seasons has been included in the Supplementary Material. Also, several references to those results have been added to the manuscript. Thanks for this comment.

**Figure 7** is excellent. It clearly shows the strong role of NAO controlling precipitation trends. Two suggestions to improve this plot. Remove RC analysis and include the magnitude of the trends from 1952 and include a new plot with correlation between WWI and WP (in x axis) relating with the magnitude of change in WWI (in y axis).

- We have created the plots required by the reviewer; however, they were not included in the manuscript. We think that it does not offer more information than the actual. Indeed, poorer Adj. R2 was found with trend quantifications. The next figure exemplifies this finding.

[Figure]

**542**. This cannot by supported by the several literature available on this issue. Humid basins in Spain are increasing vegetation coverage and activity. In Figure A5 there are some catchments in the Pyrenees and the Cantabrian chains that show negative trends in

the study but that several studies based on field information (not on land cover change modelling) that show strong revegetation process. See e.g. García-Ruiz et al. 2015 vol 170 Pirineos.

- We have clarified in the manuscript that our results regarding the forest cover changes are limited by the uncertainties of the dataset. We cannot discuss if the HILDA dataset includes mistakes in the modelling of forest cover evolution in the northern Spain. However, we would like to mention that the validation of the model had several areas in this region (see Fuchs et al., 2015, Fig. 3). We assume that the HILDA dataset could have temporal deviations of the extension of forest in Spain, while we only study the period from 1950 to 2010. Precisely, we have checked that the extension of forest cover in this region was performance from 1900 to 1950, see next plot.

[Figure]

- Meanwhile, the study period of this contribution does not register the extension of the forest cover in the northern Spain (in the same magnitude of the former), see next figure.

[Figure]

**New Forest Cover from 1950 to 1980**

**Section 4.** Reinterpretation of the "abrupt changes" is strongly necessary. This should be interpreted as short-term trends in the frame of the strong interannual and decadal variability of precipitation that characterises this region.

- Thanks for the comment, it has been reinterpreted in the full text of the manuscript.

**Answer to the Reviewer 2 comments**

The authors use many different terms, and not all of them are consistent and properly defined. While the main focus is on streamflow, several other terms and variables are used, and their difference is not explained. Eq 1, for instance, uses RR and I, but their relation to streamflow is unclear. Without further explanation, I fail to see the added value of Eq. 1 and I suggest to remove it since it does not seem to have any role in the analysis. I also question whether this equation is correct in the first place: it is clearly only valid at long temporal scales (long enough for dS/dt to become zero, although this conditions is not mentioned) and large spatial scales, but exactly at these scales the distinction between (surface?) run-off and infiltration becomes a bit artificial. I suggest the authors to remove this equation 1 and 2 (for the same reason – it doesn't contribute to the analysis), and check the manuscript for consistency in terminology.

- Both equations have been removed in order to please the comments of reviewers.

In my initial review I commented on the quality and origin of the data not being clear. While the authors now refer to a text in the appendix, this text does not really address my main concern. The text also suffers from some of the same problems as outlined in the previous paragraph, for instance the statement "water outflows refer to human induced water reductions in the reserve" does not explain how water inflows relate to water outflows (the focus of the dataset) not streamflow (the focus of the study). What exactly was measured? Water levels in the basins? Water level in the stream? At which location with respect to the reservoir? Using what techniques? What is the potential impact of assumptions made in the streamflow estimation on the results? How were water levels translated to flows? None of this is explained, while I consider this to be vital information as much of the value of the study lies in the fact that observed streamflow is used, rather than a derived or modelled product with its own caveats.

- We think that this issue has been resolved. Whereas more details of the procedures computing the water inflows estimation have been included in the Supplementary Material, the terminology has been revised in the manuscript. Right, we refer to streamflow in our results for several times, meanwhile we worked with water inflows to reservoirs. Thanks for this comment. In the final version of the

manuscript, we do not mention streamflow considering our results (the data used in the study).